# A Bibliometric Analysis of Stroke Caregiver Research from 1989 to 2022

**DOI:** 10.3390/ijerph20054642

**Published:** 2023-03-06

**Authors:** Mohd Azmi Bin Suliman, Tengku Muhammad Hanis, Mohd Khairul Anwar Kamdi, Mohd Ismail Ibrahim, Kamarul Imran Musa

**Affiliations:** Department of Community Medicine, School of Medical Sciences, Universiti Sains Malaysia, Kubang Kerian 16150, Malaysia

**Keywords:** bibliometric analysis, stroke caregivers, research trends, research evaluation studies

## Abstract

Many stroke survivors suffer with varying degrees of disability and require assistance. Family members commonly act as informal caregivers, caring for these stroke survivors and ensuring care adherence. However, many caregivers reported a poor quality of life and physical and psychological distress. Due to these issues, multiple studies have been conducted to understand the experience of caregivers, the outcomes of caregiving, and interventional studies among caregivers. This study aims to explore the intellectual landscape of studies on stroke caregivers using bibliometric analysis. Studies with “stroke” and “caregiver” terms in the title were extracted from the Web of Sciences (WOS) database. The resulting publications were analysed using the ‘*bibliometrix*’ package in R. There were 678 publications analysed, dating from 1989 to 2022. The USA has the highest number of publications (28.6%), followed by China (12.1%) and Canada (6.1%). The most productive institution, journal and author were The University of Toronto (9.5%), ‘Topics in Stroke Rehabilitation’ journal (5.8%) and Tamilyn Bakas (3.1%), respectively. Co-occurrences keywords analysis revealed mainstream research on stroke survivors, burden, quality of life, depression, care, and rehabilitation, reflecting the timeless hotspot in the field. This bibliometric analysis helps us understand the current state of stroke caregiver research and its recent developments. This study can be used to evaluate research policies and promote international cooperation.

## 1. Introduction

Stroke is a significantly debilitating disease, and its frequency is anticipated to increase worldwide [1]. Despite improving mortality and morbidity rates among stroke patients, stroke survivors may acquire residual impairment. Acute stroke episodes affect stroke patients and people close to them. In addition, stroke survivors also often experience considerable distress. Many family members reported they did not anticipate the stroke attack, and most were unprepared for the consequences [2].

As stroke changes the lives of stroke patients and those close to them, stroke episodes bring new expectations and roles, such as the caregiver. Those close to the stroke patient adapt to this new role of caregiver, giving physical, emotional, economic and spiritual support and trying their best to fulfil the stroke survivor’s needs [3,4]. To adapt to these changes, caregivers need to equip themselves with appropriate knowledge, skills and abilities which are essential for the stroke survivors’ care and the caregiver’s well-being. The rehabilitation team, especially the stroke physician, rehabilitation nurse, occupational therapist and physiotherapist, play essential roles in helping the caregivers to gain new knowledge. Healthcare workers must see caregivers as integral to stroke care to ensure stroke survivors adhere to care plans and therefore get closer to their pre-stroke condition [5].

Unfortunately, caring for a relative with morbidity can place strain upon caregivers. Stroke caregivers often report developing varying degrees of physical and psychiatric problems. Some caregivers develop tiredness, insomnia, anxiety and depression, which lead to poor quality of life [6,7]. Thus, stroke care plans should focus on stroke survivors and include caregivers as an integral part of treatment regimes. In addition, stroke survivors and caregivers dyad approaches should be adapted and tailored to the needs of stroke survivors and their caregivers [5,8].

In this sense, the caregiver is the focus of attention, which justifies the development of this research. Given the overall burden of stroke and the importance of stroke caregivers, it is unsurprising that there has been considerable research in this field. There are spectra of journals that cover this field, ranging from broad topic journals such as “BMJ Open”, to more specific topic journals such as “Topic in Stroke Rehabilitation” and “Stroke”.

Bibliometrics analysis has been around for several decades. Bibliometric analysis has recently gained traction with the increased availability of databases and software [9,10]. Bibliometric analysis is a statistical method with two main techniques: (1) performance analysis, which is to measure the production of scientific research and trends, and (2) scientific mapping, which is to examine the relationship of intellectual interactions and structural connections between intellectual constituents. Bibliometric analysis is crucial to map scientific knowledge and establish nuances from the large volume of data and metadata of scientific contributions in respective fields [11]. The analysis may reveal underrepresented areas for generating research opportunities and scientific development. Bibliometric analysis also has been applied to analyse journal publications and compare different journals [12].

While recent studies have covered many aspects of the topic, limited academic publications have tried to understand the research pattern related to stroke caregivers systematically. There were several bibliometric studies on post-stroke care [13] and general caregivers [14], but to the best of our knowledge, this is the first bibliometric study on stroke caregivers. Such an analysis will help delineate the global research trend related to caregivers’ health and the interventions available. This analysis may also help the readers to identify research gaps, especially in underrepresented regions and communities. Researchers and institutions can also use this study’s findings as a benchmark for their research directions and policies.

The main aim of this study is to explore research patterns, specifically (1) identifying influential authors, (2) finding journals that were most represented in studies, (3) finding collaboration patterns between countries, authors, and institutions, and (4) finding significant keywords or hotspots related to stroke caregivers.

## 2. Materials and Methods

### 2.1. Study Design, Search Terms and Study Flow

This study is a bibliometric analysis. First, the criteria for selection were established. To ensure that the publications were relevant to stroke caregivers, we searched publication titles with the term “stroke” and “caregiver”. The list of publications was extracted from the Web of Science Core Collection database from the Web of Science (WOS), which includes Social Sciences Citation Index (SSCI), Science Citation Index Expanded (SCI-EXPANDED) and Emerging Sources Citation Index (ESCI). The search term used in the search equation was “(TI = (stroke) AND TI = (caregiver))”, in which we did not limit the time range. The search was performed on 6 December 2022. The selection criteria were limited to articles, reviews, or proceed papers. After the search, BibTeX data were downloaded, containing the details of the publication, including author names, titles, journal names, author keywords, publication years, cited references and abstracts. Two independent researchers validated the search and data extraction.

### 2.2. Statistical Analysis

There were several analyses performed in this study. The first analysis was a descriptive analysis in which we attempted to quantify total numbers and the top ten for language, country, institution, journal, author, article and keywords related to the field. We also mapped (1) a collaboration network map between countries and institutions, (2) a co-citation network map between journals and authors, and (3) a co-occurrences network map for keywords.

Two parameters were used to identify the most influential publications in the field: the most-cited articles and the highest average citations per year. The decision to use these two parameters was made because most-cited articles may favour older articles; therefore, an adjustment was made to include more recent publications [12].

In identifying the most influential journals in the field, Bradford’s law of scattering was used in this study. Bradford’s law of scattering states that scientific journals can be arranged by productivity and grouped into three zones of an approximately equal number of total publications. Zone 1, or the core zone, is a small group of journals that produce about 33.3% of total publications; Zone 2 is a larger group of journals that make up approximately another 33.3% of total publications; and Zone 3 is the biggest group of journals, which produces the remaining 33.3% of total publications [15]. Therefore, it can be understood that in a specific field, a few core journals produce most of the publications, while most journals only produce a few.

In ensuring the network mapping remained legible and clear, the network maps were limited to the top 25 items (i.e., the top 25 most productive journals, authors, and most common keywords), except for collaboration between countries and institutions. For collaboration networks between institutions, the network maps were limited to the top 10% of institutions, and the names of institutions were not included to avoid overcrowding the plot. For the same reason, so as not to overcrowd the plot, for the collaboration network between countries, countries with no collaboration with other countries were excluded from the plot. Since there was no definite guidance on the number of items to include in the network maps, the number of items was chosen arbitrarily.

The bibliometric analysis was performed using the R (version 4.2.2) within the RStudio (version 2022.07.2) [16,17]. Packages used for the bibliometric analysis include ‘*tidyverse*’ and ‘*bibliometrix*’ packages [9,18].

### 2.3. Ethical Considerations

Ethical review was not required for this study as this study was conducted without any human subjects.

## 3. Results

### 3.1. General Information

There were 1006 publications that contained the terms “stroke” and “caregiver” in the title; however, only 678 publications fulfilled the inclusion and exclusion criteria. The flow of the publication search process is shown in Figure 1.

Among the total 678 publications analysed, the publications were produced between 1989 and 2022. There were more articles (91.7%) than review articles (8.3%). The publications were authored by 2456 authors in 260 journals, and used 1085 keywords (after de-duplication). The total number of citations was 14,749. The number of publications produced annually has fluctuated but generally increased since 2004, with an annual growth rate of 11.4%, as shown in Figure 2.

### 3.2. Influential Articles

The top ten publications based on the number of citations and the average number of citations per year are shown in Table 1 and Table 2, respectively. Only one publication among the top 10 highly cited publications was produced in the past ten years. Among highly cited publications, most of the publications were quantitative longitudinal studies measuring the quality of life, well-being, and burden of caregivers. When adjusted to the year of publication, the top 10 average citations per year included more recent publications. The publications achieving the top 10 average citations per year were a mixture of quantitative and qualitative research, observational and interventional studies, and review articles.

### 3.3. Language

The publications were published in nine languages, with most of them in English, at 96.0%, followed by German (1.0%), Portuguese (0.9%), Spanish (0.7%), French (0.6%), Korean (0.3%); the rest were Italian, Russian and Turkish, with 0.1% each.

### 3.4. Authors

Out of the 678 publications, the number of authors per publication ranged from one author to 27 authors, within which publications with four authors were the most common (21.1%), as shown in Figure 3.

There were 2456 authors in the field, most appearing once (81.4%). As shown in Table 3, Tamilyn Bakas of the USA was the most prolific researcher, with 21 publications.

When mapping the bibliographic coupling of the 25 most productive authors, there were five main clusters, all of which were intercorrelated, as shown in Figure 4.

### 3.5. Author’s Keyword and WOS’s Keywords-Plus

Among the 678 publications, there were 1085 author’s keywords and 923 WOS keywords-plus. Apart from the term “stroke” and “caregiver”, other common keywords used include “survivors”, “burden”, “quality of life”, “depression”, “care”, and “rehabilitation”. The top 10 keywords for both authors and WOS’s keywords-plus are shown in Table 4.

When the top 25 author’s keyword co-occurrences were mapped, there were three main clusters, all of which were correlated. The authors’ keyword co-occurrences network map is shown in Figure 5.

### 3.6. Journals

A total of 260 journals have been published in this field. Based on Bradford’s law of scattering, 12 core journals produced 33.0% of total publications, as in Table 5. There were 54 journals in Zone 2 and 194 journals in Zone 3. The top three journals were “Topics In Stroke Rehabilitation” (n = 40), with a 2021 journal impact factor (JIF) of 2.18; “Stroke” (n = 21), with a 2021 JIF of 10.17; and “Rehabilitation Nursing” (n = 21), with a 2021 JIF of 1.46. In Zone 1 of Bradford’s law, there was a mixture of general medical journals and specialist journals related to stroke, rehabilitation and nursing.

The 25 most productive journals were mapped for a co-citation network, in which there was only one cluster and all the journals were correlated, as shown in Figure 6.

### 3.7. Institutions

The publications in this field were from 1042 institutions, with the University of Toronto having the highest number of publications. The top 10 productive institutions are shown in Table 6.

Among the top 10% of the institutions, there were 11 clusters of collaboration. Most of the clusters were intercorrelated, as shown in Figure 7. The names of the institutions were not included, to avoid overcrowding plot. Refer to Appendix A for the labels for each node.

### 3.8. Countries

Up to 20.1% of the 678 publications had international co-authorships. When analysing the corresponding author’s country, 678 publications were published in 51 countries. The USA has the highest number of publications (28.6%) and total citations (38.4%), as shown in Table 7.

The 51 countries were grouped based on the World Bank’s country income groups (refer to Appendix A), which found that high-income countries had the highest number of publications (68.1%), as shown in Table 8.

Among all the corresponding author’s countries, there were three main clusters of collaboration, with the main clusters consisting of countries with a significant contribution to the field, as shown in Figure 8.

## 4. Discussion

We performed a simple search within the available large body of literature. This bibliometric analysis demonstrates the nature and range of scientific literature on stroke caregivers. This bibliometric analysis can be considered as a surrogate to identify targeted journals and seek international co-authors in the field. To our knowledge, this is the first study to systematically describe influential publications, authors and journals at the international level for studies related to stroke caregivers. There were several key findings worth discussing here, including (1) trends of studies conducted; (2) common keywords used by the authors; (3) collaborations between authors, institutions, and countries; and (4) the importance of these key findings to policymakers and health care providers.

Overall, publications on stroke caregivers have been on the rise since the concept of family caregivers emerged in the 2000s, and especially in the last ten years [35,36]. In our study, we noticed that many influential publications were quantitative longitudinal studies trying to understand caregiving experiences and problems arising coincidentally. However, when adjusted to the year of publication, the top 10 publications by average citations per year included more recent publications. Furthermore, the type of studies in the top 10 by average citations per year also varied, including (1) qualitative studies, trying to understand the issues in depth; (2) interventional studies, both focusing on the caregiver and the caregiver–survivor dyad; and (3) review papers, in which many authors were trying to synthesise the various findings of previous studies. This trend suggests that globally, researchers are interested in multi-faceted issues related to stroke caregivers, and in trying to tackle or improve these issues [37]. However, when trying to understand the most cited articles, the readers need to be aware that the analysis may favour older articles [12].

In our study, almost all the studies were in English, as expected. This should not be surprising, as English is the global lingua franca, even for science and technology. While publishing in English may not be easy for non-English speakers, publications in English are more accessible to worldwide readers [38].

Stroke caregivers’ concepts are multidimensional, and among them, this study identified several keywords commonly used by authors. In the literature, in addition to the term “stroke” and “caregiver”, other related terms such as “survivor”, “burden”, “quality of life”, “depression”, “family”, and “care” were closely related to stroke caregivers. These terms reflect (1) the key player in stroke caregiving (i.e., the stroke caregiver, stroke survivors and their family members), (2) the typical issues that stroke caregivers face (e.g., burden, quality of life and depression), and (3) their needs (e.g., stroke care, stroke rehabilitation and the health of both stroke survivors and caregivers). We, however, wanted to caution the reader that since in this bibliometric analysis we include both original articles and reviews, the keywords may be inflated when they appeared in both original articles and reviews that may consist of the aforementioned original articles. This was expected, but we can consider the keywords influential and important if they repeatedly appeared both in original articles and reviews [10,11].

Our study identified the core journals according to Bradford’s law of scattering to indicate the most influential journals. As mentioned previously, the core journals in this field were a mixture of general medical journals and specialist journals, including stroke-related journals, rehabilitation-related journals, and nursing-related journals. In addition, all the journals in the core journals group were high-impact journals. Choosing peer-reviewed journals with high impact is vital to ensure that the findings of the studies are valid, thus guaranteeing the quality of the evidence [39]. This is important because many policymakers and healthcare providers depend on high-quality evidence [40]. Many authors also consider several factors in selecting the journals they want to publish, such as high visibility journals, which are indicated by high impact, open access, database indexation, a swift review process, a high acceptance rate and low fees [41].

When looking at the corresponding authors’ countries of origin, high-income countries such as the United States, Canada and China predominantly contributed to publications. Coincidently, most of the most influential institutions in this field also primarily come from the North American region, with some mixture with institutions from Europe and Asia, suggesting that most studies were carried out in high-income countries. The findings reflect general trends previously identified in stroke caregivers research [37]. From another perspective, many middle-income and low-income countries had higher stroke burdens [42], and the number of people with stroke needing care has increased rapidly in developing countries [42]. The GBD 2019 Stroke Collaborator reported that the mortality rate and stroke-related disability-adjusted life expectancy among low-income countries were 3.6 and 3.7 times worse, respectively, than high-income countries [43]. Still, our study found that corresponding authors from low-income countries were severely underrepresented. Unfortunately, the lack of publications among middle-income and low-income countries may suggest that these countries faced more significant challenges and threats because of inadequate service support [44].

Upon further investigation, the countries’ collaboration network maps in our studies show three main clusters, which may indicate mixed signs. On the one hand, we applaud good international collaboration, as illustrated by the largest cluster in our study. Studies from different countries and institutions were intercorrelated with each other. Some of the studies on middle-income and low-income countries were authored by researchers from institutions in high-income countries. For example, Yan et al. (2016) consisted of a team of authors from China, the USA, Peru and other countries, being affiliated with high-income countries, but the article discussed stroke care in low- and middle-income countries [45]. Publishing in the common lingua franca also encourages international collaboration [38]. Unfortunately, on the other hand, there was also a lack of cooperation in some other countries. Variation in healthcare systems, cultures, family structures and care arrangements across countries and regions can lead to different health problems and countermeasures for caregivers, which may be reasons for the inconsistency in international cooperation in this field of research [34,46,47]. To navigate these global healthcare issues, professionals in public health should collaborate.

Our study also makes clear that the different publications have covered a broad spectrum of knowledge in the field, which includes (1) understanding the caregivers’ experience; (2) the complications, and the determinants of the complications; and (3) interventions that target not only stroke survivors, but target stroke caregivers too. Furthermore, given the increasing reliance on caregivers in healthcare, research on stroke caregivers has become more important than ever.

In addition, several research gaps were identified in this study, including a lack of participation of researchers from low-income countries. Researchers and institutions may use the findings of this study as guidelines for future research, including research direction and policy.

### Strength and Limitations

Within this data-driven research, there are several methodological benefits: (1) the procedure is transparent and reproducible; (2) this study is scalable, meaning researchers can adjust the boundaries of data features accordingly; and (3) the quantitative method of measuring the research’s impact and academic performance is reliable [48].

However, one of the limitations of this study concerns the search terms. By limiting the search terms to the publication titles, we can ensure that the publications analysed fulfil inclusion criteria. However, we might miss relevant publications that might not contain both “stroke” and “caregiver” terms in the title. These publications were excluded because they were not well catalogued and were not published according to our inclusion criteria.

In addition, this bibliometric study was performed in a manner akin to a cross-sectional study, meaning we did not examine the trend of the studies related to stroke caregivers. By comparing analyses at several time points, we may identify dynamic and thematic changes over time [10]. However, this was beyond the scope of this manuscript, and we would recommend further studies to explore trends.

Apart from WOS, several databases such as Scopus and PubMed can provide information for bibliometric analysis. Each database has its own data collection policy which affects the scope of the publication and the number of citations. However, inconsistencies in the citation indexes and metadata structures make it challenging to include all the databases in a single bibliometric analysis. Other bibliometric research has also been known to have this limitation [49].

Nonetheless, we would like to stress that our bibliometric research was different from that of other review articles, as review articles emphasise search strategies, the eligibility of studies and the risk of bias evaluation, whereas bibliometric research instead provides a bigger picture of trends and topic areas.

## 5. Conclusions

Thus far, we have reviewed, analysed and discussed articles on stroke caregivers published in peer-reviewed journals between 1989 and 2022. The number of stroke caregiver studies has increased recently, and scholars from various countries are collaborating to achieve their academic goals. Understanding of current research trends outlines the knowledge map in the research field and helps to find gaps that require further study. While we identified that most studies on stroke caregivers were concentrated in high-income countries, there is a need for further research on stroke caregivers, especially in low-income and middle-income countries, as they were found to be more in need of a care support system for stroke caregivers.

Finally, researchers specialising in stroke caregivers can set benchmarks for themselves and network with peers by using the resources analysed within this bibliometric analysis.

## Figures and Tables

**Figure 1 ijerph-20-04642-f001:**
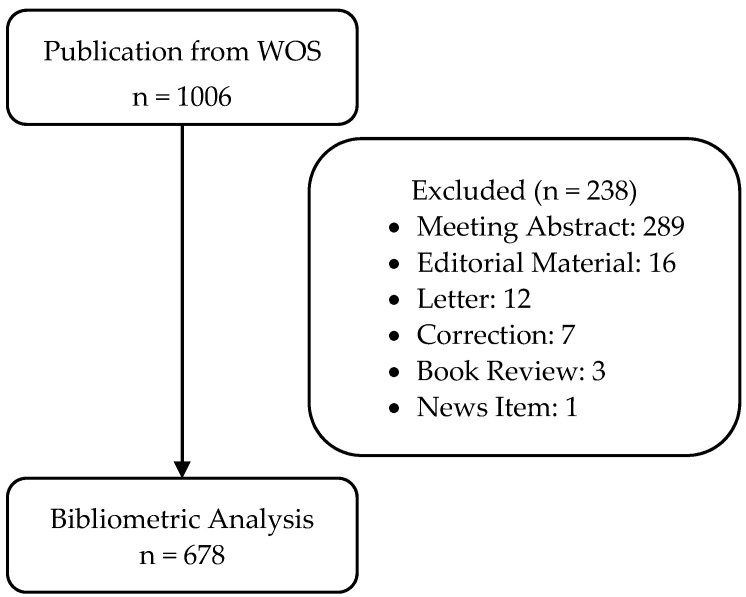
Publication search process flow.

**Figure 2 ijerph-20-04642-f002:**
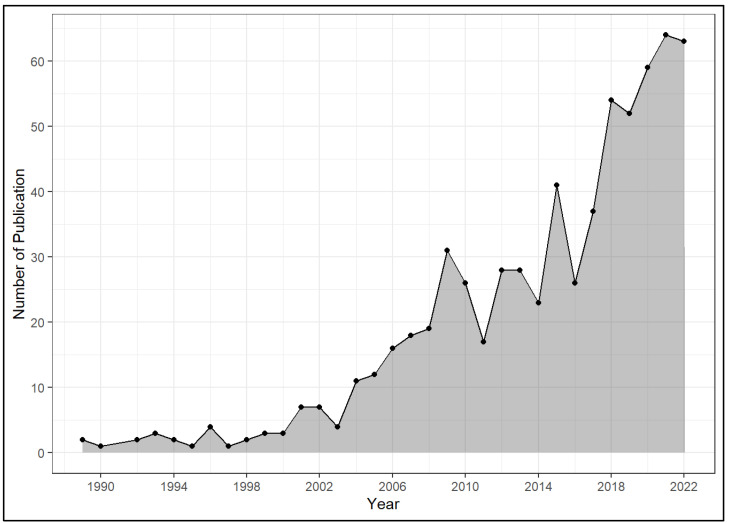
Annual scientific production (n = 678).

**Figure 3 ijerph-20-04642-f003:**
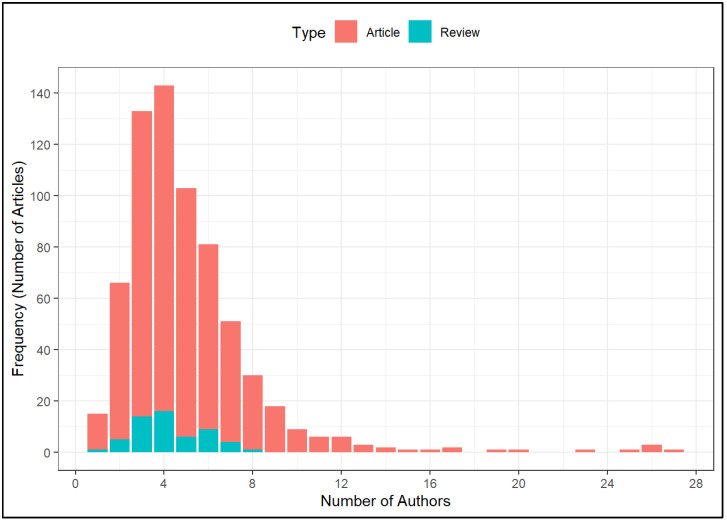
Number of authors per publication (n = 678).

**Figure 4 ijerph-20-04642-f004:**
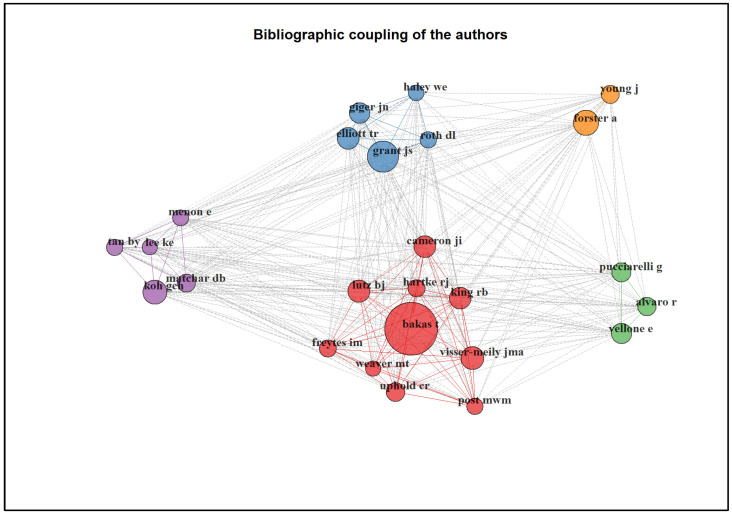
Co-citation network map between authors (n = 25).

**Figure 5 ijerph-20-04642-f005:**
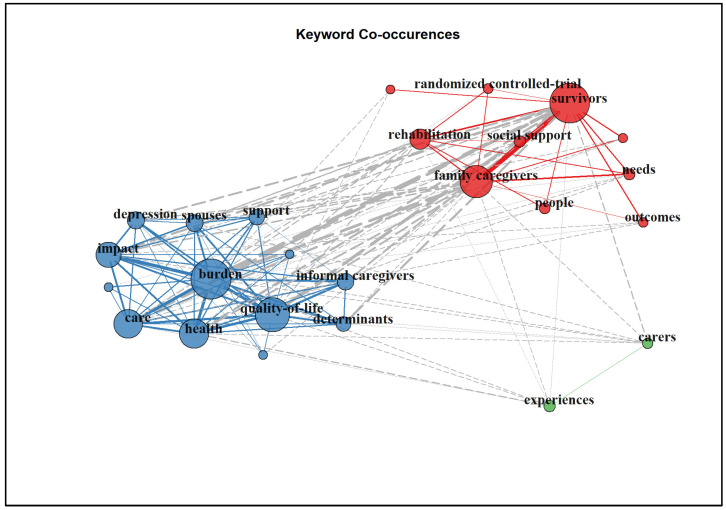
Author’s keywords co-occurrence network map (n = 25).

**Figure 6 ijerph-20-04642-f006:**
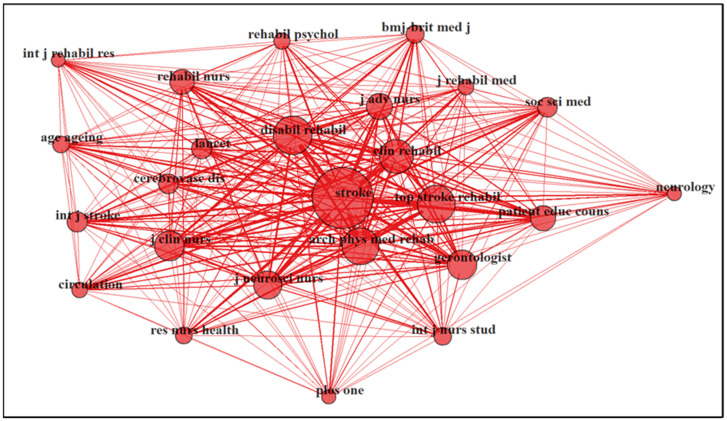
Co-citation network map between journals (n = 25).

**Figure 7 ijerph-20-04642-f007:**
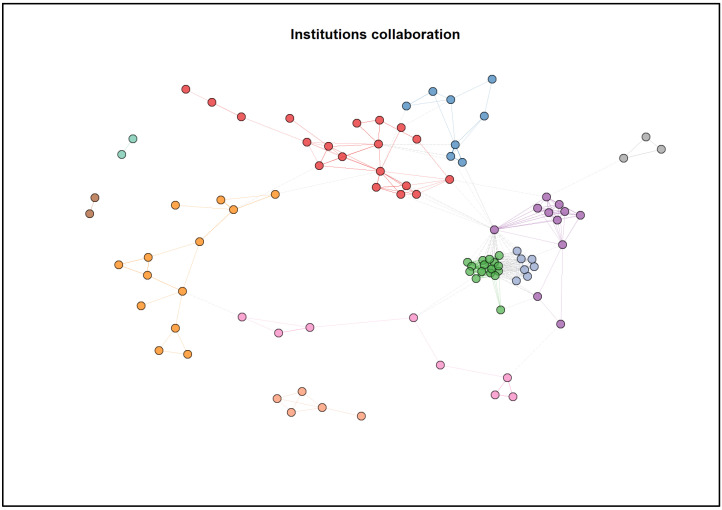
Collaboration network map between institutions ^1^: cluster 1—red, cluster 2—dark blue, cluster 3—orange, cluster 4—purple, cluster 5—light blue, cluster 6—green, cluster 7—grey, cluster 8—pink, cluster 9—light brown, cluster 10—dark brown and cluster 11—light green (n = 104).^1^ Refer to Appendix A for labels for each node.

**Figure 8 ijerph-20-04642-f008:**
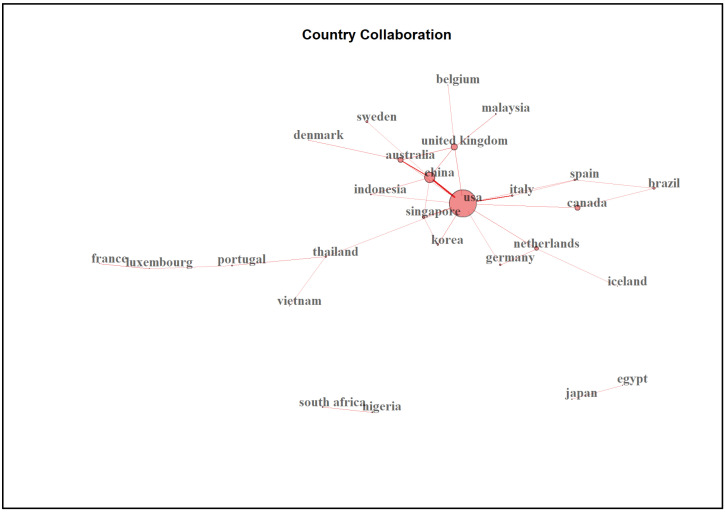
Collaboration network map between countries (n = 27).

**Table 1 ijerph-20-04642-t001:** Top 10 articles with the most citations.

Rank	Title	Author, Year	Number of Citations	Average Number of Citations per Year
1	Determinants Of Caregiving Burden and Quality of Life in Caregivers of Stroke Patients	McCullagh et al., 2005 [19]	277	15.39
2	Determinants Of Quality of Life in Stroke Survivors and Their Informal Caregivers	Jönsson et al., 2005 [20]	220	12.22
3	Caregiver’s Burden of Patients 3 Years After Stroke Assessed by A Novel Caregiver Burden Scale	Elmståhl, Malmberg and Annerstedt, 1996 [21]	209	7.74
4	Telephone Intervention with Family Caregivers of Stroke Survivors After Rehabilitation	Grant et al., 2002 [22]	204	9.71
5	A Systematic Review of Caregiver Burden Following Stroke	Rigby, Gubitz and Phillips, 2009 [23]	192	13.71
6	Top 10 Research Priorities Relating to Life After Stroke-Consensus from Stroke Survivors, Caregivers, and Health Professionals	Pollock et al., 2014 [24]	190	21.11
7	Stroke Patients’ Informal Caregivers-Patient, Caregiver, and Service Factors That Affect Caregiver Strain	Bugge, Alexander and Hagen, 1999 [25]	183	7.63
8	Caregiver Burden and Health-Related Quality of Life Among Japanese Stroke Caregivers	Morimoto, Schreiner and Asano, 2003 [26]	180	9.00
9	A Comparison of Caregivers for Elderly Stroke and Dementia Victims	Draper et al., 1992 [27]	177	5.71
10	“Timing It Right”: A Conceptual Framework for Addressing the Support Needs of Family Caregivers to Stroke Survivors From The Hospital To The Home	Cameron and Gignac, 2008 [28]	170	11.33

**Table 2 ijerph-20-04642-t002:** Top 10 articles by average citations per year.

Rank	Title	Author, Year	Number of Citations	Average Number of Citations per Year
1	The Global Prevalence of Anxiety and Depressive Symptoms Among Caregivers of Stroke Survivors	Loh et al., 2017 [29]	131	21.83
2	Top 10 Research Priorities Relating to Life After Stroke-Consensus from Stroke Survivors, Caregivers, Additionally, Health Professionals	Pollock et al., 2014 [24]	190	21.11
3	Stroke Survivors’ and Informal Caregivers’ Experiences of Primary Care and Community Healthcare Services-A Systematic Review and Meta Ethnography	Pindus et al., 2018 [30]	82	16.40
4	Evidence For Stroke Family Caregiver and Dyad Interventions A Statement for Healthcare Professionals from The American Heart Association and American Stroke Association	Bakas et al., 2014 [31]	143	15.89
5	Determinants Of Caregiving Burden and Quality of Life in Caregivers of Stroke Patients	McCullagh et al., 2005 [19]	277	15.39
6	Poststroke Spasticity Sequelae and Burden on Stroke Survivors and Caregivers	Zorowitz, Gillard and Brainin, 2013 [32]	140	14.00
7	A Systematic Review of Caregiver Burden Following Stroke	Rigby, Gubitz and Phillips, 2009 [23]	192	13.71
8	A Structured Training Programme for Caregivers of Inpatients After Stroke (TRACS): A Cluster Randomised Controlled Trial and Cost-Effectiveness Analysis	Forster et al., 2013 [33]	85	12.64
9	The influence of Chinese culture on family caregivers of stroke survivors: A qualitative study	Qiu, Sit and Koo, 2018 [34]	60	12.60
10	Determinants Of Quality of Life in Stroke Survivors and Their Informal Caregivers	Jönsson et al., 2005 [20]	220	12.22

**Table 3 ijerph-20-04642-t003:** The most productive authors (n = 2486).

Rank	Author Name	Number of Publications, n (%)
1	Tamilyn Bakas	21 (3.1)
2	Joan S. Grant	15 (2.2)
3	Barbara J Lutz	13 (1.9)
=4	David L Roth	12 (1.8)
=4	Ercole Vellone	12 (1.8)
=6	William E Haley	11 (1.6)
=6	Gerald C H Koh	11 (1.6)
=6	Gianluca Pucciarelli	11 (1.6)
=9	Rosario Alvaro	10 (1.5)
=9	Jill I Cameron	10 (1.5)
=9	Amy Forster	10 (1.5)
=9	Linda L Pierce	10 (1.5)
=9	Victoria Steiner	10 (1.5)

**Table 4 ijerph-20-04642-t004:** Top 10 most relevant keywords: author’s keywords and WOS’s keyword-plus.

Rank	Author’s Keyword	Publications, n (%)	WOS’s Keyword-Plus	Publications, n (%)
1	Stroke	453 (66.8)	Survivors	181 (26.6)
2	Caregivers	198 (29.2)	Burden	179 (26.4)
3	Caregiver	118 (17.4)	Quality-of-life	149 (21.9)
4	Depression	82 (12.0)	Family caregivers	137 (20.2)
5	Quality Of Life	73 (10.7)	Care	136 (20.0)
6	Rehabilitation	70 (10.3)	Health	132 (19.4)
7	Burden	44 (6.4)	Impact	114 (16.8)
8	Family Caregivers	44 (6.4)	Rehabilitation	88 (12.9)
9	Caregiver Burden	38 (5.6)	Depression	75 (11.0)
10	Anxiety	30 (4.4)	Spouses	75 (11.0)

**Table 5 ijerph-20-04642-t005:** List of journals in Zone 1 of Bradford’s law of scattering (n = 678).

Rank	Journal Name	Number of Publications, n (%)
1	Topics In Stroke Rehabilitation	40 (5.8)
2	Stroke	29 (4.2)
3	Rehabilitation Nursing	21 (3.0)
4	Journal of Neuroscience Nursing	20 (2.9)
5	Journal of Clinical Nursing	18 (2.6)
6	Journal of Advanced Nursing	17 (2.5)
7	Disability and Rehabilitation	16 (2.3)
8	Rehabilitation Psychology	15 (2.2)
9	BMJ Open	14 (2.0)
10	Clinical Rehabilitation	13 (1.9)
11	Archives of Physical Medicine and Rehabilitation	11 (1.6)
12	Scandinavian Journal of Caring Sciences	10 (1.4)

**Table 6 ijerph-20-04642-t006:** Most productive institution (n = 1042).

Rank	Institution	Number of Publications, n (%)
1	University of Toronto	65 (9.5)
2	University of Florida	60 (8.8)
3	The University of Alabama at Birmingham	49 (7.2)
4	National University of Singapore	38 (5.6)
5	The University of Texas Health Science Center at Houston	36 (5.3)
6	University of Cincinnati	34 (5.0)
7	Chinese University of Hong Kong	31 (4.5)
8	Maastricht University	26 (3.8)
=9	Università degli studi di Roma Tor Vergata	26 (3.8)
=9	Zhengzhou University	26 (3.8)

**Table 7 ijerph-20-04642-t007:** Top 10 most productive countries and number of total citations by country.

Highest Number of Publications			Highest Number of Total Citations			
Rank	Country	Number of Publications, n (%)	Rank	Country	Total Citations	Average Article Citations
1	USA	193 (28.6)	1	USA	4720	24.46
2	China	82 (12.1)	2	United Kingdom	1472	50.76
3	Canada	41 (6.1)	3	Canada	1455	35.49
4	Netherlands	33 (4.9)	4	China	1240	15.12
5	United Kingdom	29 (4.3)	5	Netherlands	1183	35.85
6	Brazil	24 (3.6)	6	Sweden	737	49.13
7	Australia	23 (3.4)	7	Australia	590	25.65
8	Turkey	21 (3.1)	8	Korea	319	17.72
=9	Korea	18 (2.7)	9	Italy	289	19.27
=9	Singapore	18 (2.7)	10	Denmark	277	277.00

**Table 8 ijerph-20-04642-t008:** Top 10 most productive countries and number of total citations by the World Bank’s country income groups.

World Bank Country Income Group	Number of Publication, n (%)	Total Citations	Average Article Citations
High Income	462 (68.1)	12,421	26.89
Upper-Middle Income	172 (25.4)	1796	10.44
Lower-Middle Income	39 (5.6)	354	9.08
Low Income	2 (0.3)	11	5.50

## Data Availability

The BibTeX data can be publicly accessed via https://zenodo.org/record/7421945 (accessed on 11 January 2023).

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
