# Peer review of "A Bibliometric Analysis of Stroke Caregiver Research from 1989 to 2022"

_ijerph, 2023, doi:10.3390/ijerph20054642_

Round 1
Reviewer 1 Report
In this manuscript, the authors perform an analysis of articles with the words “stroke” and “caregiver” terms in the title from the Web of Sciences database. While the work seems to be performed correctly, I don't feel like it sheds much light either on the underlying topic of stroke caregivers, nor on the networks of scientific collaborations. The lack of quantitative analysis beside summary statistics, or another topic to make comparisons with, significantly limits what inferences can be gleaned from this paper. Something like figure 4 might be of more interest if combined with some kind of network analysis or at least some other graphs to compare side-by-side. Similarly, figure 5 of keyword cooccurrences is not put in context of what it might mean. The clustering of institutional collaborations (fig 7) might be interesting if presented with more information or an explanatory theory about the relationships and their structure. In summary, the paper would need significant revisions in order to increase the value to readers of the journal in term of conveying novel information or testing some hypothesis.
Author Response
Point 1: In this manuscript, the authors perform an analysis of articles with the words “stroke” and “caregiver” terms in the title from the Web of Sciences database. While the work seems to be performed correctly, I don't feel like it sheds much light either on the underlying topic of stroke caregivers, nor on the networks of scientific collaborations.
Response 1: Thank you for the comment. We appreciate the concern; however, we want to reiterate that bibliometric study is unique, that while it may not offer deep insight into the topic, but it offers a broad overview of the subject (refer (Zupic and ÄŒater, 2015) and (Donthu et al., 2021)). Therefore, we hope our manuscript may offer the reader the width of the theme related to stroke caregivers. In addition, to our best knowledge, our manuscript is the first bibliometric study on the specific topic of stroke caregivers. Thus, while some information offered in our manuscript might seem obvious, this was the first to describe the information, which justifies this manuscript.
We update the manuscript at lines 71-73 to add to the justification of this manuscript.
Point 2: The lack of quantitative analysis beside summary statistics, or another topic to make comparisons with, significantly limits what inferences can be gleaned from this paper. Something like figure 4 might be of more interest if combined with some kind of network analysis or at least some other graphs to compare side-by-side. Similarly, figure 5 of keyword cooccurrences is not put in context of what it might mean.
Response 2: Thank you for the comment. We acknowledge that there were underlying mathematical calculations (and resulting statistical parameters) such as for network analysis, as described by (Zupic and ÄŒater, 2015; Donthu et al., 2021). However, in our humble opinion, the network analysis can adequately explain with the accompanying graph. Furthermore, several recent similar bibliographic studies (refer (Devos and Menard, 2019; Akmal et al., 2020; Sun et al., 2022) on the different topics had minimal statistical analysis besides summary statistics.
We also agree that it would be interesting if we could compare, for example, between different periods, as described by (Zupic and ÄŒater, 2015); however, we want to limit our bibliometric study to what we already explained in our study. Therefore, we add this as a limitation, as written in lines 339-343.
Point 3: The clustering of institutional collaborations (fig 7) might be interesting if presented with more information or an explanatory theory about the relationships and their structure. In summary, the paper would need significant revisions in order to increase the value to readers of the journal in term of conveying novel information or testing some hypothesis.
Response 3: Thank you for the comment. While we acknowledge that it will be more interesting, it may require more time to update the manuscript substantially. We have updated several points (including from other reviewers); however, we cannot prioritise these points. We wholeheartedly apologise that we are unable to update the manuscript on this point.
Donthu, N. et al. (2021) ‘How to conduct a bibliometric analysis: An overview and guidelines’, Journal of Business Research, 133, pp. 285–296. Available at: https://doi.org/10.1016/j.jbusres.2021.04.070.
Zupic, I. and ÄŒater, T. (2015) ‘Bibliometric Methods in Management and Organisation’, Organizational Research Methods, 18(3), pp. 429–472. Available at: https://doi.org/10.1177/1094428114562629.
Akmal, M. et al. (2020) ‘Glioblastome Multiforme: A Bibliometric Analysis’, World Neurosurgery, 136, pp. 270–282. Available at: https://doi.org/10.1016/j.wneu.2020.01.027.
Devos, P. and Menard, J. (2019) ‘Bibliometric analysis of research relating to hypertension reported over the period 1997-2016’, Journal of Hypertension, 37(11), pp. 2116–2122. Available at: https://doi.org/10.1097/HJH.0000000000002143.
Sun, H.-L. et al. (2022) ‘Schizophrenia and Inflammation Research: A Bibliometric Analysis’, Frontiers in Immunology, 13, p. 907851. Available at: https://doi.org/10.3389/fimmu.2022.907851.
Reviewer 2 Report
1. In the last paragraph of the Introduction, the authors state the aims of the current literature bibliometric analyses. These aims include identifying influential authors and their collaborators, identifying journal connected to stroke caregivers, and/or identifying keyword ‘hotspots’ related to the caregivers of stroke survivors. The authors’ Introduction does not indicate ‘Why’ these things are important to learn.
2. The authors write bibliometric analyses ‘reveal emerging areas for scientific development’ or possibilities for ‘generating new research opportunities’. It is not clear how these types of results match the aims for the study identified in the last paragraph of the Introduction (see #1 above) as to what one can expect from a bibliometric analysis.
3. . The Introduction to the manuscript lists six journals (paragraph four) that cover research articles covering the caregiver of stroke patients. Only three of these journals are listed as being important in the Results. The internal inconsistency between the Introduction and the Results is not addressed.
4. Inclusion of reviews in the analyses (review articles are 8.3% of the included dataset) may have resulted in some terms/authors/etc. being counted multiple times. Similarly, the most heavily cited articles may have a disproportionate influence on the results (some words from a single experiment counted in the analyses multiple times). These things can be avoided by only including primary research articles in the bibliometric analyses. That is, one suggestion is to add review articles to the list of excluded items.
5. There is no evaluation for the quality of the evidence in this search. In other types of literature searches, especially those related to clinical medicine, one would expect to see an evaluation that weighs well-conducted studies more heavily than less-well conducted studies. An evaluation of the words used alone considers all studies on a topic to be equally valid. In addition, a research paper that contains several experiments is evaluated in the described bibliometric analyses as containing the same information as a single case study. Reliance on the journal impact factor alone (Discussion paragraph 5) is not a substitute for the quality of the experimental investigation reported.
6. The inclusion of a network maps for citations, authors, and keywords relating to the topic is rather interesting. The clustering of certain aspects of each is a unique approach to making connections.
7. The number of authors per paper analyses and figure (Figure 3) can be removed. It is difficult to determine what this adds to the main point of this manuscript.
8. The analyses of the ‘Most Productive Institutions’ is difficult to justify. People do research. Institutions do not do research. People also move from one institution to others during their careers. Similarly, people collaborate, the institutions do not collaborate. More thorough justifications are needed for inclusion of these parts of the analyses (maybe remove Table 6 and Figure 7?).
9. The first paragraph of the Discussion concludes by stating the findings of this bibliometric analyses are important for policy makers and health care providers. Evidence to support this connection is lacking. Indeed, how does this conclusion match what the authors say bibliometric analyses can achieve in the Introduction?
10. The sixth paragraph of the Discussion (continued into paragraph 7) addresses the observation that higher income countries contribute to the majority of the studies examined in the bibliometric analysis. No consideration is given to the possibility that in higher income countries more people who have strokes survive (to need long term caregivers) as a result of access to advances in healthcare or that more peer reviewed research takes place in higher income countries in most areas of healthcare investigation. An interesting question arises as to a possible association between the proportion of people who have strokes and the number of research articles on stroke caregivers in different countries.
11. . There is no basis from the data presented to support the suppositions in the eighth paragraph of the Discussion that the current research ‘makes clear the different publications MUST (emphasis added) cover a broad spectrum of knowledge in the field.’. However, the bibliometric analysis conducted does not speak at all to what should be done. It only collects information on what has been done based on words in manuscripts. Similarly, the analysis conducted indicates nothing about possible interventions for the caregivers of stroke survivors.
12. The three paragraphs in the Practical Implications portion of the Discussion says things, that may be true, are beyond the scope of the bibliometric analyses conducted. The data collected do not address the things stated. The things noted are distractions from what the data reveal. These paragraphs can be removed without compromising the main points of the collected data.
13. Overall, this is an interesting approach to learning about a research area. It does not address the details or quality of research being published. The bibliometric analyses approach does address connections between countries, journals, investigators, and the English words used in research publications.
Author Response
Point 1. In the last paragraph of the Introduction, the authors state the aims of the current literature bibliometric analyses. These aims include identifying influential authors and their collaborators, identifying journal connected to stroke caregivers, and/or identifying keyword ‘hotspots’ related to the caregivers of stroke survivors. The authors’ Introduction does not indicate ‘Why’ these things are important to learn.
Response 1: Thank you for the comment. Apart from the justification listed in lines 63-74, we also add several lines to justify the ‘why’ questions in lines 74-77.
Point 2. The authors write bibliometric analyses ‘reveal emerging areas for scientific development’ or possibilities for ‘generating new research opportunities’. It is not clear how these types of results match the aims for the study identified in the last paragraph of the Introduction (see #1 above) as to what one can expect from a bibliometric analysis.
Response 2: Thank you for your concern. We have rephrased the sentence accordingly. Refer to lines 47-77
Point 3. The Introduction to the manuscript lists six journals (paragraph four) that cover research articles covering the caregiver of stroke patients. Only three of these journals are listed as being important in the Results. The internal inconsistency between the Introduction and the Results is not addressed.
Response 3: Thanks for the comment. We take note of that matter and make the changes accordingly. Refer to lines 55-57
Point 4. Inclusion of reviews in the analyses (review articles are 8.3% of the included dataset) may have resulted in some terms/authors/etc. being counted multiple times. Similarly, the most heavily cited articles may have a disproportionate influence on the results (some words from a single experiment counted in the analyses multiple times). These things can be avoided by only including primary research articles in the bibliometric analyses. That is, one suggestion is to add review articles to the list of excluded items.
Response 4: Thank you for your comment. We understand the concern; however, it was expected that certain keywords and authors might be counted multiple times, as bibliometric study concerns academic production and bibliography information. Based on the bibliometric study, any keywords or authors that were counted numerous times were considered important or influential. Several similar bibliometric studies also include both original articles and review articles, including the study by (Devos and Menard, 2019; Akmal et al., 2020; Sun et al., 2022).
We respectfully decline to exclude the review articles. However, we add lines 270-275 to highlight your concern.
Point 5. There is no evaluation for the quality of the evidence in this search. In other types of literature searches, especially those related to clinical medicine, one would expect to see an evaluation that weighs well-conducted studies more heavily than less-well conducted studies. An evaluation of the words used alone considers all studies on a topic to be equally valid. In addition, a research paper that contains several experiments is evaluated in the described bibliometric analyses as containing the same information as a single case study. Reliance on the journal impact factor alone (Discussion paragraph 5) is not a substitute for the quality of the experimental investigation reported.
Response 5: Thank you for the comment. We understand the concern. This is, however, the nature of the bibliometric analysis, in which the study measures the productivity of research publication and bibliography information rather than the content of the publication included in the analysis (Zupic and ÄŒater, 2015). The advantage of bibliometric analysis is that bibliometric study is a quantitative analysis of the academic production as a proxy for research conducted. We try to make sense of the large volumes of unconstructed data in rigorous ways (Donthu et al., 2021). The data may include hundreds or thousands of publications compared to reviews or meta-analyses. Thus bibliometric analysis quantifies the width of the study conducted related to the topic (i.e. stroke caregiver) rather than the depth.
We acknowledge the comment; however, no adjustments on the manuscript related to this point.
Point 6. The inclusion of a network maps for citations, authors, and keywords relating to the topic is rather interesting. The clustering of certain aspects of each is a unique approach to making connections.
Response 6: Thank you for the kind compliment.
Point 7. The number of authors per paper analyses and figure (Figure 3) can be removed. It is difficult to determine what this adds to the main point of this manuscript.
Response 7: Thank you for the comment. We would like to argue that the number of authors per paper analyses and Figure 3 were part of descriptive analyses of bibliometric study (Zupic and ÄŒater, 2015); thus, there were no changes in this point.
Point 8. The analyses of the ‘Most Productive Institutions’ is difficult to justify. People do research. Institutions do not do research. People also move from one institution to others during their careers. Similarly, people collaborate, the institutions do not collaborate. More thorough justifications are needed for inclusion of these parts of the analyses (maybe remove Table 6 and Figure 7?).
Response 8: We agree with your point. We believe that while people do research, the institution more or less influences the policy and direction of the research conducted in the institution by the researchers.
Institutions (both most productive as found in the analysis or other institutions) interested in focusing on stroke caregivers' studies may refer to this manuscript as a benchmark and guide them for future direction and policy at their institution.
We have added lines 323-326 in the discussion to emphasise this explanation.
Point 9. The first paragraph of the Discussion concludes by stating the findings of this bibliometric analyses are important for policy makers and health care providers. Evidence to support this connection is lacking. Indeed, how does this conclusion match what the authors say bibliometric analyses can achieve in the Introduction?
Response 9: Thank you for the comment. We update the discussion at lines 323-326
Point 10. The sixth paragraph of the Discussion (continued into paragraph 7) addresses the observation that higher income countries contribute to the majority of the studies examined in the bibliometric analysis. No consideration is given to the possibility that in higher income countries more people who have strokes survive (to need long term caregivers) as a result of access to advances in healthcare or that more peer reviewed research takes place in higher income countries in most areas of healthcare investigation. An interesting question arises as to a possible association between the proportion of people who have strokes and the number of research articles on stroke caregivers in different countries.
Response 10: Thank you for the interesting view. We added a new table in both the main body of the article and in the supplementary material, in which we grouped the corresponding author by income group based on World Bank classification (Table 8 and Table S1) and discussed the stroke burden among the group based on Global Stroke Burden report 2018 and the productivity as found in our study (Line 294-298).
Point 11. There is no basis from the data presented to support the suppositions in the eighth paragraph of the Discussion that the current research ‘makes clear the different publications MUST (emphasis added) cover a broad spectrum of knowledge in the field.’. However, the bibliometric analysis conducted does not speak at all to what should be done. It only collects information on what has been done based on words in manuscripts. Similarly, the analysis conducted indicates nothing about possible interventions for the caregivers of stroke survivors.
Response 11: Thank you for the comment. We have rephrased the text, which we initially intended to mean something else. The word ‘must’ has been changed to ‘had’.
As for possible interventions, we agree that our study doesn’t directly indicate any possible interventions, as bibliometric analysis only describes and summarise the bibliography information of previous publication. However, the most influential publication (Table 1 and Table 2) did list out several interventional studies, which readers can refer to for the intervention.
Point 12. The three paragraphs in the Practical Implications portion of the Discussion says things, that may be true, are beyond the scope of the bibliometric analyses conducted. The data collected do not address the things stated. The things noted are distractions from what the data reveal. These paragraphs can be removed without compromising the main points of the collected data.
Response 12: Thank you for the comment. We acknowledge the concern and omit the discussion.
Point 13. Overall, this is an interesting approach to learning about a research area. It does not address the details or quality of research being published. The bibliometric analyses approach does address connections between countries, journals, investigators, and the English words used in research publications.
Response 13: Thank you for the kind compliment. We agree with what the reviewer has commented.
Donthu, N. et al. (2021) ‘How to conduct a bibliometric analysis: An overview and guidelines’, Journal of Business Research, 133, pp. 285–296. Available at: https://doi.org/10.1016/j.jbusres.2021.04.070.
Zupic, I. and ÄŒater, T. (2015) ‘Bibliometric Methods in Management and Organization’, Organizational Research Methods, 18(3), pp. 429–472. Available at: https://doi.org/10.1177/1094428114562629.
Akmal, M. et al. (2020) ‘Glioblastome Multiforme: A Bibliometric Analysis’, World Neurosurgery, 136, pp. 270–282. Available at: https://doi.org/10.1016/j.wneu.2020.01.027.
Devos, P. and Menard, J. (2019) ‘Bibliometric analysis of research relating to hypertension reported over the period 1997-2016’, Journal of Hypertension, 37(11), pp. 2116–2122. Available at: https://doi.org/10.1097/HJH.0000000000002143.
Sun, H.-L. et al. (2022) ‘Schizophrenia and Inflammation Research: A Bibliometric Analysis’, Frontiers in Immunology, 13, p. 907851. Available at: https://doi.org/10.3389/fimmu.2022.907851.
Round 2
Reviewer 1 Report
Thank you for your direct responses.
Author Response
Comment Reviewer 1
Point 1: Thank you for your direct responses.
Response 1: You are most welcome. We also would like to express our gratitude for the constructive comments.